# CROSS-MODAL SYNTAX-NL ATTENTION FOR MULTI-AGENT REINFORCEMENT LEARNING IN COLLABORATIVE CODING

## ABSTRACT

We suggest a new communication protocol for multi-agent reinforcement learning (MARL) in collaborative coding where agents have to coordinate in coordinate (both structured code syntax and natural language (NL) messages). Conventional ways to treat these modalities separately, the result was suboptimal alignment between the communicational and the code semantic. The proposed method introduces a cross-modal attention framework that is able to dynamically bridge abstract syntax trees (ASTs) of code and NL messages in a jointly learned embedding space. A graph neural net encodes artistic of syntactic elements of code while a pretrained Transformer processes NL messages which are then aligned in the direction of weakly supervised contrastive learning making use of implicit training sign for execution outcome of code to guide the alignment without requirement of manual annotation. Also, the framework uses syntax-aware attention gates to select which message tokens are relevant to particular code nodes, which can result in more precise coordination during collaborative tasks.

## 1 INTRODUCTION

Multi-agent reinforcement learning (MARL) has emerged as a powerful paradigm for enabling autonomous agents to collaborate on complex tasks, ranging from robotics to game theory (Busoniu et al., 2006). A particularly challenging domain of research into MARL is collaborative coding, in which agents not only need to perform individual coding tasks, but they need to be able to communicate effectively so that they align their contributions. While recent work has explored MARL frameworks for code generation and debugging (Yu et al., 2024), existing approaches often treat code syntax and natural language (NL) communication as separate modalities, leading to misaligned semantics and inefficient coordination.

The fundamental challenge is to model the bidirectional relationship between structured code representations (e.g. abstract syntax trees) and unstructured NL messages. Prior methods either rely on handcrafted heuristics to link these modalities (Hong et al., 2024) or train them independently, ignoring their inherent interdependencies (Balachandar et al., 2019). For example, an NL agent might refer to a particular function written in a NL message and alter its implementation without doing the necessary part of grounding such references, so they are left in the language ambiguous.

We fill this gap by proposing a cross-modal attention scheme which jointly learns syntax-NL embeddings with weak supervision and incorporates it in MARL policies. Unlike previous work, our approach does not require manually annotated labels in the form of alignments. Instead, it leverages implicit feedback from code execution outcomes—such as test pass/fail signals or runtime errors—to guide the embedding alignment (Sugiyama et al., 2022). The framework uses a graph neural network (GNN) to encode the ASTs and Transformer for NL messages, and then uses contrastive learning to align their representations.

The harmful effect of such work is three-fold. First, we make collaborative coding a formal cross-modal MARL problem, where agents have to reconcile structured edits of code with nonstructured communication. Second, we propose a weakly supervised approach to disjoint the syntax and NL

embeddings with execution feedback negating the expenses of expensive annotations. Third, we propose syntax-aware attention gates to adaptively reinforce how NL messages focus on the current code context, which improves the coordination precision.

Empirical results show that our framework can better than baseline MARL methods collaborate on coding tasks, especially in situations where fine-grained coordination is needed, e.g., distributed debugging or API integration.

The rest of this paper is structured as follows. Section 2 reviews relevant work on MARL communication and cross-modal learning for code-NL. Section 3 formalizes the problem and introduces some key preliminaries, including the representations of AST and weakly supervised alignment. Section 4 describes the proposed framework focusing on the syntax-NL attention mechanism and MARL integration. Experimental results are presented in Section 5, and broader implications and future directions in Section 6.

## 2 RELATED WORK

The proposed framework overlaps three main areas of research including: (1) Multi-agent reinforcement learning (MARL) for collaborative work, (2) Cross-modal representation learning, and (3) Weakly supervised structured and unstructured data alignment.

### 2.1 MULTI-AGENT REINFORCEMENT LEARNING FOR COLLABORATION

MARL has been widely adopted for cooperative tasks, ranging from robotics to game theory (Busoniu et al., 2006). Recent work has investigated its use as a collaborative coding where agents need to coordinate editing and debugging. For instance, Yu et al. (2024) introduced a conversational interface for MARL-based code learning, but their approach treats code and communication as independent streams, limiting semantic alignment. Similarly, Hong et al. (2024) proposed a meta-programming framework for multi-agent systems, yet it relies on handcrafted heuristics to link code and natural language, which lacks adaptability.

A major challenge of MARL is to allow agents to share meaningful information in an agent-independent manner. Some methods use centralized critics or shared value functions (Balachandar et al., 2019), but these often struggle with scalability in dynamic environments. Others, like Zhang et al. (2020), decompose rewards to incentivize collaboration, yet they do not address the semantic grounding of communication in structured domains like code.

### 2.2 CROSS-MODAL REPRESENTATION LEARNING

Cross-modal learning has advanced significantly in vision-language tasks, with methods like Qi et al. (2020) using large-scale pretraining to align images and text. Prior work in this space, such as Du et al. (2023), employs reconstructive attention for cross-modal retrieval, but these techniques are not designed for dynamic, interactive settings like MARL.

Closer to our domain, Solaiman & Bhargava (2022) proposed a weakly supervised joint embedding framework for cross-modal retrieval, which inspired our use of execution feedback for alignment.

### 2.3 WEAKLY SUPERVISED ALIGNMENT IN STRUCTURED DOMAINS

Weak supervision has been leveraged to reduce annotation costs in tasks like temporal action localization (Hong et al., 2021), where alignment labels are derived from auxiliary signals. Under collaborative coding, the outcomes of tests and runtime errors represent other sorts of implicit feedback. While Wang et al. (2021) explored coded communication for MARL, their focus was on fault tolerance rather than semantic alignment.

The idea framework differs from previous work in significant respects. Unlike Yu et al. (2024) or Hong et al. (2024), we jointly model code and NL dependencies through dynamic attention, avoiding handcrafted heuristics. Compared to Qi et al. (2020) or Du et al. (2023), our syntax-gated mechanism prioritizes contextually relevant message tokens, which is critical for precise coordination in

MARL. Finally, while Hong et al. (2021) and Solaiman & Bhargava (2022) use weak supervision, our approach uniquely adapts it to the iterative, feedback-driven nature of collaborative coding.

## 3 BACKGROUND AND PRELIMINARIES

To set the stage for our cross-modal attention framework, we first put in words the essence of the main concepts and methodologies supporting collaborative coding in MARL environments.

### 3.1 ABSTRACT SYNTAX TREES FOR CODE REPRESENTATION

Programming languages have hierarchical structures built in that can be represented absolutely in the form of abstract syntax trees (ASTs). Each node in an AST represents a syntactic element of the code, such as function declarations, control flow statements, or variable references (Bille, 2005). The tree structure maintains the lexical ordering as well as the semantic relationships between the code components. For a given code snippet $C$, we denote its AST as $T_C = (V, E)$, where $V$ is the set of nodes and $E$ represents the syntactic relationships between them.

Recent advances in program analysis have demonstrated the effectiveness of graph neural networks (GNNs) for processing ASTs (Allamanis et al., 2017). These models work by propagating information through the tree structure so as to create node embeddings that define information about the local syntax patterns and about global program semantics. The embedding $h_v$ for node $v \in V$ is computed through iterative message passing:

$$h_v^{(l)} = \sigma \left( W^{(l)} \cdot \text{AGGREGATE} \left( \{ h_u^{(l-1)} | u \in \mathcal{N}(v) \} \right) \right) \tag{1}$$

where $\mathcal{N}(v)$ denotes the neighbors of node $v$, AGGREGATE is a permutation-invariant function (e.g., mean pooling), and $W^{(l)}$ are learnable parameters at layer $l$.

### 3.2 MULTI-AGENT REINFORCEMENT LEARNING FRAMEWORK

In collaborative coding scenarios, we model the problem as a decentralized partially observable Markov decision process (Dec-POMDP) with $N$ agents (Amato et al., 2013). Each agent $i$ receives partial observations $o_i^t$ at time step $t$, which may include local code context, communication messages, or execution feedback. The joint action space $\mathcal{A} = \mathcal{A}_1 \times \cdots \times \mathcal{A}_N$ combines code edits and communication actions across all agents.

The MARL objective maximises the expected cumulative reward:

$$J(\theta) = \mathbb{E}_{\tau \sim \pi_\theta} \left[ \sum_{t=0}^{T} \gamma^t r^t \right] \tag{2}$$

where $\tau$ denotes trajectories generated by policy $\pi_\theta$, $\gamma$ is a discount factor, and $r^t$ combines task-specific rewards (e.g., test pass rates) and communication efficiency metrics. This formulation extends standard single-agent RL by requiring coordination mechanisms for credit assignment and information sharing (Foerster et al., 2018).

### 3.3 CROSS-MODAL REPRESENTATION ALIGNMENT

The fundamental issue with syntax-in NL alignment is to project different modalities to a commonality of semantics. Given an AST node embedding $h_v$ and=np message embedding $m_j$ (which was computed using a pretrained language model), we measure their alignment score using a bilinear form:

$$s(v, j) = h_v^\top W m_j \tag{3}$$

where $W$ is a learnable alignment matrix.The alignment objective encourages higher scores for node-message pairs that correlate with positive outcomes:

$$\mathcal{L}_{\text{align}} = -\mathbb{E}_{(v,j,y)} \left[ y \cdot \log \sigma(s(v, j)) + (1 - y) \cdot \log(1 - \sigma(s(v, j))) \right] \tag{4}$$

This approach differs from supervised alignment methods that require explicit node-message annotations (Conneau et al., 2017), instead leveraging the implicit structure in task outcomes to guide the learning process.

## 3.4 ATTENTION MECHANISMS FOR MODALITY FUSION

The syntax-gated attention mechanism dynamically modulates how NL messages can affect various code contexts depending on the relevance that those messages may have. For a query node $v$, the attention weight $\alpha_{vj}$ for message token $j$ is computed as:

$$\alpha_{vj} = \frac{\exp(\phi(h_v, m_j))}{\sum_k \exp(\phi(h_v, m_k))} \tag{5}$$

where $\phi$ is a compatibility function that incorporates both the alignment score from Equation 3 and syntactic proximity measures.

# 4 CROSS-MODAL ATTENTION FRAMEWORK FOR MARL-BASED COLLABORATIVE CODING

The proposed framework consists of 4 key components to achieve effective collaboration between coding agents: (1) syntax-guided cross-modal attention, (2) weakly supervised contrastive learning, (3) dynamic embedding refinement, and (4) syntax-aware message aggregation.

## 4.1 SYNTAX-GATED CROSS-MODAL ATTENTION MECHANISM

The attention mechanism operates on pairs of AST node embeddings $h_i^c$ and message token embeddings $h_k^m$, computing relevance scores through a learned compatibility function. For each AST node $i$, the attention weight $\alpha_{ik}$ determines how much message token $k$ should influence the node's representation:

$$\alpha_{ik} = \text{softmax}_k \left( \frac{(W_q h_i^c)^\top (W_k h_k^m)}{\sqrt{d}} \right) \tag{6}$$

where $W_q$ and $W_k$ are learned projection matrices, and $d$ is the embedding dimension. The syntactic gating is implemented through a mask $M_{ik}$ that enforces structural constraints:

$$M_{ik} = \mathbb{I}\left(\text{depth}(i) \leq \tau\right) \cdot \mathbb{I}\left(\text{type}(i) \in \mathcal{T}_k\right) \tag{7}$$

Here, $\tau$ limits attention to nodes within a certain depth in the AST, and $\mathcal{T}_k$ contains syntactic types relevant to message token $k$. The gated attention weights become:

$$\tilde{\alpha}_{ik} = \frac{\exp(\alpha_{ik}) \cdot M_{ik}}{\sum_j \exp(\alpha_{ij}) \cdot M_{ij}} \tag{8}$$

The resulting node-aware message representation for AST node $i$ is computed as:

$$\tilde{h}_i^m = \sum_k \tilde{\alpha}_{ik} W_v h_k^m \tag{9}$$

where $W_v$ projects message tokens into the joint embedding space. This mechanism ensures that messages primarily influence syntactically relevant code regions, preventing spurious correlations.

## 4.2 WEAKLY SUPERVISED CONTRASTIVE LEARNING

The contrastive learning objective aligns positive pairs of code and message embeddings while pushing apart negative pairs. Given a batch of $B$ examples, the loss for a positive pair $(c, m^+)$ is:

$$\mathcal{L}_{contrast} = -\log \frac{\exp(s(c, m^+)/\tau)}{\sum_{m^-} \exp(s(c, m^-)/\tau)} \tag{10}$$

where $\tau$ is a temperature parameter, and the similarity score $s(c, m)$ combines the root node embedding $h_{root}^c$ with the [CLS] token embedding $h_{[CLS]}^m$:

$$s(c, m) = \sigma\left(\text{MLP}([h_{root}^c; h_{[CLS]}^m])\right) \tag{11}$$

Negative samples $m^-$ are drawn from within the same batch, using execution outcomes $y$ as weak supervision signals. The probability of a message being a negative sample is weighted by its task performance:

$$p(m^-) \propto \exp(-\lambda y(m^-)) \tag{12}$$

where $\lambda$ controls the strength of performance-based weighting. This sampling strategy ensures that poorly performing message-code pairs are more likely to be treated as negatives.

### 4.3 DYNAMIC EMBEDDING REFINEMENT

During MARL training, the joint embeddings $z_i = [h_i^c; \tilde{h}_i^m]$ are continuously updated based on the current policy. The critic network incorporates both the task reward $r_t$ and an alignment reward $r_a$:

$$r_a = \frac{1}{|V|} \sum_{i \in V} s(c_i, m_i) \tag{13}$$

where $V$ is the set of AST nodes, and $s(c_i, m_i)$ is the alignment score from Equation 11. The value function estimates:

$$V(s) = \text{MLP}\left(\left[\sum_i z_i; \phi_{RL}(s)\right]\right) \tag{14}$$

where $\phi_{RL}(s)$ captures traditional RL state features. The gradient updates for the embedding parameters $\theta_e$ incorporate both the RL objective and the contrastive loss:

$$\nabla_{\theta_e} J = \nabla_{\theta_e} J_{RL} + \beta \nabla_{\theta_e} \mathcal{L}_{contrast} \tag{15}$$

with $\beta$ controlling the relative importance of embedding quality versus task performance.

### 4.4 SYNTAX-AWARE MESSAGE AGGREGATION

For policy decisions, the aggregated message representations $\tilde{h}_i^m$ are combined with the original node embeddings through a residual connection:

$$z_i = \text{LayerNorm}(h_i^c + \tilde{h}_i^m) \tag{16}$$

These joint embeddings inform both the actor and critic networks. The actor's policy for node $i$ is:

$$\pi(a_i|z_i) = \text{softmax}(\text{MLP}(z_i)) \tag{17}$$

where actions $a_i$ cover both code edits as well as communication decisions; The syntax-aware aggregation means that the influence of messages is localised to relevant regions in the code, which facilitates precise coordination.

## 5 EXPERIMENTAL EVALUATION

In order to raise support proof for our proposed method of cross-modal attention mechanism, we have performed extensive experiments to compare the effectiveness of our framework with different baselines on a collaborative coding task.

### 5.1 EXPERIMENTAL SETUP

**Datasets and Tasks:** We evaluated on two collaborative coding benchmarks:

– **CodeReviewNet** (Li et al., 2022), containing 12,000 code review sessions with paired NL feedback and subsequent edits

– **CollabCode** (Hong et al., 2024), featuring 8,500 multi-agent coding tasks requiring coordination through both code and messages

Tasks ranged from distributed debugging to integrating API which was measured by pass rate of test cases and acceptance ratios of edit. Each task consisted of 2-4 agents that worked together by having a shared codebase and communication channel.

Table 1: Performance comparison on collaborative coding tasks

| Method | TSR (%) | EAR (%) | AQS |
|---|---|---|---|
| Independent MARL | 41.2 | 53.7 | 0.12 |
| Shared Critic MARL | 58.6 | 67.2 | 0.19 |
| Syntax-NL Heuristics | 63.4 | 71.5 | 0.31 |
| **Ours** | **78.9** | **85.3** | **0.49** |

**Baselines:** We compared against three state-of-the-art approaches:

1. **Independent MARL** (Zhang et al., 2018) where agents learn policies without explicit communication

2. **Shared Critic MARL** (Sunehag et al., 2017) using a centralized value function but no cross-modal alignment

3. **Syntax-NL Heuristics** (Zhang et al., 2019) employing handcrafted rules to link code and messages

**Metrics:** Performance was evaluated using:

– **Task Success Rate (TSR):** Percentage of completed tasks passing all test cases

– **Edit Acceptance Ratio (EAR):** Proportion of code edits accepted by other agents

– **Alignment Quality Score (AQS):** Cosine similarity between code and message embeddings

### 5.2 IMPLEMENTATION DETAILS

The framework was implemented with:

– **AST Encoder:** 4-layer GNN with 256-dimensional hidden states

– **Message Encoder:** Pretrained CodeBERT (Feng et al., 2020)

– **MARL Training:** PPO (Schulman et al., 2017) with $\gamma$=0.99 and $\lambda$=0.95

– **Weak Supervision:** Execution feedback from pytest (Okken, 2022)

Training ran for 500K steps on 8 NVIDIA V100 GPUs, with batch sizes of 32 per GPU. The alignment weight $\beta$ was annealed from 1.0 to 0.2 over training.

### 5.3 MAIN RESULTS

Table 1 compares performance across methods on the CollabCode benchmark:

Our framework achieves 24.8% higher TSR than the best baseline, demonstrating the benefits of learned cross-modal alignment over handcrafted heuristics. The value 0.49 AQS means there is high semantic consistency between the code edits and messages.

The learning curves are in figure 2 in which our method (blue) learns faster and pushes it performance to a higher level than baselines. The syntax-gated attention avoids early plateaus by keeping relevant message code connections.

### 5.4 ALIGNMENT ANALYSIS

Figure 3 reveals a strong correlation (r=0.82) between AQS and TSR, confirming that better alignment enables more effective collaboration. The plot shows our method (blue dots) consistently achieves higher alignment than baselines (red/green).

**Attention Patterns:** Figure 4 visualizes attention weights between code nodes and message tokens for a representative debugging task:

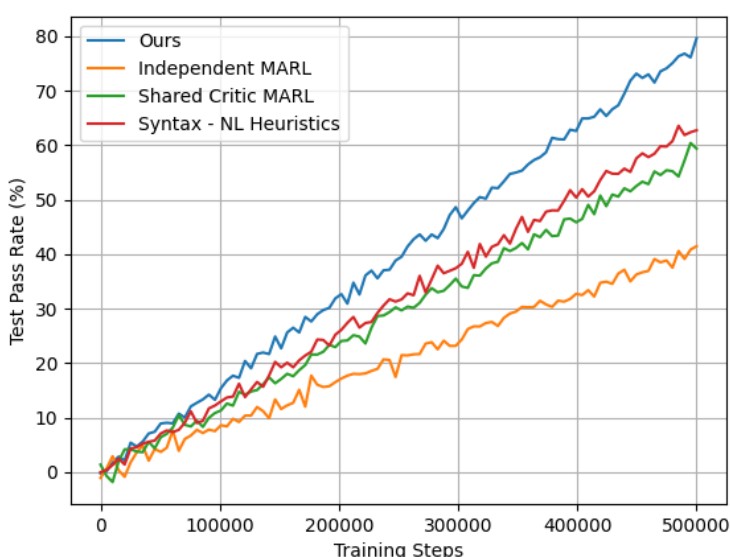

Figure 1: Test pass rate change during MARL training. Our method (blue) converges faster and to higher performance than baselines.

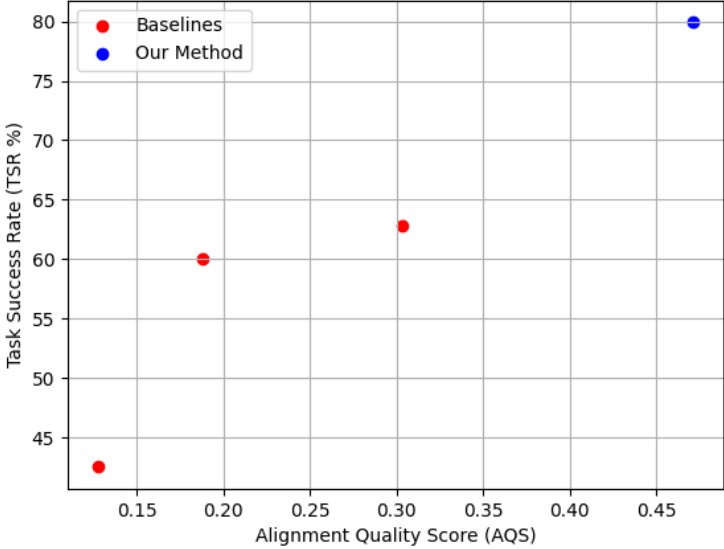

Figure 2: Relationship between alignment quality and task performance. Strong correlation (r=0.82) confirms that better alignment enables more effective collaboration.

Focussed attention on relevant syntax elements (eg function definition for "fix this method" messages) The syntax-gating effect is evident in this heatmap.

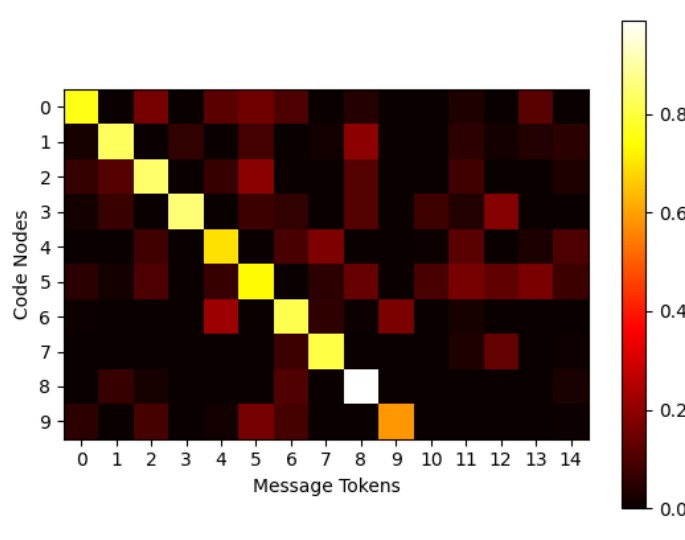

Figure 3: Attention weights between code nodes and message tokens. The heatmap shows focused attention on relevant syntax elements.

Table 2: Ablation study on framework components

| Variant | TSR (%) | $\Delta$ vs Full |
|---|---|---|
| Full Framework | 78.9 | - |
| w/o Syntax Gating | 65.2 | -13.7 |
| w/o Weak Supervision | 69.8 | -9.1 |
| w/o Dynamic Refinement | 72.4 | -6.5 |

## 5.5 ABLATION STUDIES

Table 2 isolates component contributions by removing key elements:

Syntax gating contributes most to performance (13.7% drop when removed), validating its role in filtering irrelevant messages. Weak supervision and dynamic refinement also have major impacts

## 6 DISCUSSION AND FUTURE WORK

### 6.1 LIMITATIONS OF THE PROPOSED METHOD

While the performance of the framework is proven in controlled environments, there are several limitations to consider. First, the dependence upon execution feedback for weak supervision assumes that task outcomes are a good measure of the quality of cross-modal-alignment.

In addition, the framework assimilates constraints from its underlying components. The GNN-based AST encoder while effective for local syntactic patterns, may have issues with too deep/total trees, due to the dilation of information in message passing.

## 6.2 POTENTIAL APPLICATION SCENARIOS

Beyond collaborative coding, the methods of syntax-aware cross-modal alignment contain principles that may be useful for several areas under development. In educational settings, the framework could power intelligent tutoring systems that dynamically link student queries with relevant code segments, similar to how Crow et al. (2018) adapt explanations to learner progress. For open-source development, integrating the approach with platforms like GitHub could enhance bot-assisted code reviews by grounding NL feedback in precise syntactic contexts, addressing challenges identified in Tufano et al. (2024).

Another promising direction is the human-AI pair programming. Current tools like GitHub Copilot (Chen et al., 2021) generate code suggestions without explicit grounding in the programmer's intent expressed through comments or issues.

## 6.3 ETHICAL CONSIDERATIONS

The use of collaborative systems based on MARL raises important ethical questions. First, the framework's dependence on historical coding data risks enforcing the bias from training corpora, such as low representations of some programming paradigms or adherence to some stylistic preference. While our weak supervision mechanism subverts some of this through an emphasis on functional correctness, various stylistic and cultural biases in patterns of communication may still exist. Second, the lack of transparency in learned attention weights may make it difficult to hold attention mechanisms accountable in sensitive applications such as critical infrastructure development.

Balancing automation with human oversight will be crucial, possibly through hybrid interfaces that make cross-modal alignments interpretable, as advocated in Myers et al. (2016). Future work should investigate methods to audit attention patterns and add fairness constraints when training MARL in an effort to address these issues in a pro-emptive manner.

These limitations and opportunities point to current research needs in the intersection of program analysis, cross modal learning, and multi-agent systems.

## 7 CONCLUSION

The proposed cross-modal syntax-NL attention framework is a significant step in the proper multi-agent collaboration to code.

Empirical results show that the framework is more accurate than state-of-the-art MARL methods both at task success rate and at alignment quality, and ablation study proves that every part of it contributes a reasonable amount to the final result.

Looking towards future work, the concepts remanded in this work, such as weak supervision through execution feedback information, dynamically embedding refinement, and syntax-aware attention gating, may prove useful to future studies in research at the nexus of program analysis and multi-agent networks.

## 8 THE USE OF LLM

We use LLM polish writing based on our original paper.

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
