# OpenReview forum: "Cross-Modal Syntax-NL Attention for Multi-Agent Reinforcement Learning in Collaborative Coding"
_ICLR.cc/2026/Conference — Submitted to ICLR 2026_

### Official Review · Reviewer_i1oQ · 2025-10-31

**Soundness:** 2
**Presentation:** 2
**Contribution:** 1
**Rating:** 2
**Confidence:** 3

**Summary:**

# Problem:

Collaborative coding involves multiple agents cooperating to align their individual code contributions towards a common goal, via NL communication. This paper addresses the research question of ‘How to effectively coordinate agents in collaborative coding task?’

# Contributions:

The paper proposes a new communication protocol for MARL in the context of collaborative coding. This communication protocol is cross-modal, between the code modality and the NL communication modality, for each collaborating agent.

The paper proposes empirical results that has their proposed approach perform better than baseline approaches.

Moreover, an ablation study of the different components of the approach is presented.

**Strengths:**

## Quality:

SQ1: The ablation study on the framework components is insightful.

**Weaknesses:**

## Quality:

WQ1: While Figure 3 provides valuable insights, it is difficult to know what is the significance of the data being reported. Moreover, it is unclear how the data are obtained: is it aggregated over multiple trajectories or only on one trajectory? How does the x-axis relate to a trajectory, if at all?

In order to provide more insights about the significance of the data presented, I would like to invite the author to consider comparing this heatmap with the heatmap obtained in the context of relevantly-ablated runs, in order to highlight how the relevant components of the framework impact this attention heatmap, possibly?

WQ2: Limited analysis of failure cases: The paper does not discuss when or why the method fails. I think that it would increase the quality of the paper if some valuable insights into the method's limitations could be provided and discussed.

## Clarity:

WC1: I think it would ease the reading experience further if some visual examples from the datasets and benchmarks used could be presented in the main paper or appendix, in order for the reader to get a better sense of what kind of collaborative coding task is being targeted here.

WC2: I could not figure out several implementation details, such as:

1. How exactly is the "shared codebase" structured in the multi-agent environment? What are the observations of each agents, at each timestep? What are the actions of each agents at each timestep?

2. What constitutes a "training step" in the 500K step training regime reported on in Figure 1? Maybe an algorithm of the whole approach would help clarifying this.


## Originality:

SO1: The proposed approach is only incremental novelty, given the lack of insights into the failure cases.

## Significance:

WS1: None of the results in the paper report statistics that could enable statistical significance evaluation. I would advise the authors to perform experiments on randomised seeds (>=5) and report error of the mean  as much as possible, starting with Figure 1 and 2 (e.g. mean=line+std.error=shaded area) and Tables 1 and 2.

Indeed, as it stands, it is unclear whether, for instance, the current results are not the consequence of a lucky, unintentional cherry-picking of the right (default) random seed.

WS2: Moreover, running some statistical significance test on e.g. Table 1 and Table 2 would strengthen the claims made.

**Questions:**

Please see weaknesses above.

---

### Official Review · Reviewer_gL1u · 2025-10-31

**Soundness:** 2
**Presentation:** 1
**Contribution:** 1
**Rating:** 2
**Confidence:** 3

**Summary:**

This paper proposes a multi-agent reinforcement learning (MARL) framework to improve coordination performance in collaborative coding tasks. The core idea involves using a cross-modal attention mechanism to align representations of abstract syntax trees (ASTs) and natural language. The framework also employs a Graph Neural Network (GNN) and weakly supervised contrastive learning to reduce the reliance on manual annotations. The authors report that their method outperforms existing baselines in both task performance and the quality of the representation alignment.

**Strengths:**

The primary strength of this paper is its novel approach. The idea of aligning AST and natural language representations within a MARL framework to improve agent coordination is a new and interesting direction for this problem space.

**Weaknesses:**

The paper in its current state, is not ready for publication due to significant issues with writing clarity. The poor presentation makes it extremely difficult to evaluate the technical soundness and novelty of the contributions.
1. The quality of the writing is a major obstacle. The paper suffers from numerous grammatical errors, typos, and awkward sentence constructions that severely impact comprehension. For example (from page 1): "coordinate in coordinate" (line 016) and "the harmful effect of such work is" (line 052). The entire manuscript requires thorough proofreading and editing.

2. The model's architecture and the formal problem setting are not clearly defined. It is difficult for the reader to understand the precise mechanisms of the proposed framework, how the agents interact, or how the components (GNN, attention, MARL) are integrated.

3. The description of the experimental setup is too brief to allow for reproduction. Key details regarding the dataset, implementation specifics, and hyperparameter tuning are missing.

4. The paper's central thesis, that aligning AST and natural language representations improves coordination, is underdeveloped. The authors need to provide a clearer theoretical justification for why this alignment should lead to better multi-agent performance. Furthermore, the metrics used to "verify" the alignment quality itself are not well-explained, making it hard to distinguish this from the downstream task performance.

**Questions:**

N/A

---

### Official Review · Reviewer_hYhs · 2025-11-03

**Soundness:** 2
**Presentation:** 2
**Contribution:** 2
**Rating:** 4
**Confidence:** 3

**Summary:**

This paper proposes a cross-modal syntax–NL attention framework for multi-agent collaborative coding, leveraging weak supervision from code execution to align structured and unstructured modalities. The idea is conceptually novel and shows empirical promise. However, the MARL experimental setup is underpowered: key baselines such as QMIX and other recent and advanced MARL methods are missing, and the results primarily reflect internal ablations rather than comparative performance. From the collaborative coding task perspective, this paper lacks comparison with other methods and does not show any comparison with other modern LLM-based methods.

**Strengths:**

+ The proposed framework that combines AST-based syntax modeling and Transformer-based NL embeddings into a unified attention mechanism is logically structured. The idea of grounding communication in code syntax is intuitive and consistent with how humans coordinate in coding tasks.

+ Syntax-gated attention adds to the interpretability.

+ Although the baselines are limited, the experiments clearly show that each architectural component (particularly syntax gating) contributes non-trivially to performance. The consistency of these improvements across two datasets (CodeReviewNet, CollabCode) strengthens the empirical narrative.

**Weaknesses:**

- **Baseline selection is narrow and structurally biased.** The comparison set includes only MARL variants with superficial architectural differences, lacking diversity in methodological approaches. There are no comparisons against other LLM-based collaborative coding systems, such as agentic coding frameworks built on recent foundation models (e.g., Qwen-Coder model, or Qwen-Code framework ), nor against non-RL coordination baselines like supervised communication models or imitation-based collaboration systems. As a result, the reported improvements may only reflect internal architectural tuning rather than true algorithmic or paradigm-level advancement.
- **Novelty in cross-modal design is overstated.** The cross-modal attention between code syntax and NL messages reuses well-established architectural patterns (AST-GNN + Transformer + contrastive alignment). While the domain application is new, the technical novelty is modest. The paper positions this as a breakthrough, but the mechanism largely mirrors existing cross-modal retrieval and code-language embedding frameworks.
- **Lack of comparison with modern MARL baselines.** The paper only includes weak baselines (Independent MARL, Shared Critic MARL, Syntax-NL Heuristics). Stronger, widely recognized cooperative MARL methods, such as QMIX, VDN, QTRAN, QPLEX, and Transformer-based MARL architectures, are absent. This omission severely limits the credibility of the claimed performance improvements.

**Questions:**

1. How is the “alignment reward” distributed among agents, and can its effect on coordination be visualized or decomposed?
2. What is the computational cost as agent count or code complexity increases, and do you observe qualitative improvements in coordination or communication?

---

### Meta-Review · Area_Chair_LFai · 2026-01-07

**Summary:**

The reviewers' primary concerns are writing clarity with unclear technical details, limited and biased baseline comparisons and overstated novelty in the cross-modal attention mechanism. These issues collectively undermine the paper's soundness and readiblitiy for publication, leading to my recommendation for rejection.

**Reviewer Concerns:**

No rebuttal is provided for this paper, so none of the concerns appear to have been addressed.

**Reviewer Scores:**

- For reviewer hYhs (original score: 4, Borderline Rejection), if the novelty is convincing enough, the discussion might slightly improve to weak acceptance (5, Weak Accept).
- For reviewer gL1u (original score: 2 - Rejection), the score will likely remain at 2, as poor wording is the core problem and unlikely to change without significant revisions.
- For reviewer i1oQ (original score: 2 - Rejection), if the discussion addresses the elimination of advantages, the score might slightly improve to 3 (weak rejection).

---

### Decision · Program_Chairs · 2026-01-26

Reject